# The Effect of Adipocyte-Secreted Factors in Activating Focal Adhesion Kinase-Mediated Cell Signaling Pathway towards Metastasis in Breast Cancer Cells

**DOI:** 10.3390/ijms242316605

**Published:** 2023-11-22

**Authors:** Noshin Mubtasim, Lauren Gollahon

**Affiliations:** 1Department of Biological Sciences, Texas Tech University, 2500 Broadway, Lubbock, TX 79409, USA; nmubtasim.mubtasim@ttu.edu; 2Obesity Research Institute, Texas Tech University, 2500 Broadway, Lubbock, TX 79409, USA

**Keywords:** adipocytokine, FAK, conditioned medium, breast cancer cells, non-tumorigenic breast cells, signal transduction, in vitro, 3T3-L1 MBX, obesity

## Abstract

Obesity-associated perturbations in the normal secretion of adipocytokines from white adipocytes can drive the metastatic progression of cancer. However, the association between obesity-induced changes in secretory factors of white adipocytes and subsequent transactivation of the downstream effector proteins impacting metastasis in breast cancer cells remains unclear. Focal adhesion kinase, a cytoplasmic signal transducer, regulates the biological phenomenon of metastasis by activating downstream targets such as beta-catenin and MMP9. Thus, the possible role of phosphorylated FAK in potentiating cancer cell migration was investigated. To elucidate this potential relationship, MCF7 (ER+), MDA-MB-231 (Triple Negative) breast cancer cells, and MCF-10A non-tumorigenic breast cells were exposed to in vitro murine adipocyte-conditioned medium derived from 3T3-L1 MBX cells differentiated to obesity with fatty acid supplementation. Our results show that the conditioned medium derived from these obese adipocytes enhanced motility and invasiveness of breast cancer cells. Importantly, no such changes were observed in the non-tumorigenic breast cells. Our results also show that increased FAK autophosphorylation was followed by increased expression of beta-catenin and MMP9 in the breast cancer cells when exposed to obese adipocyte-conditioned medium, but not in the MCF10A cells. These results indicate that adipocyte-derived secretory factors induced FAK activation through phosphorylation. This in turn increased breast cancer cell migration and invasion by activating its downstream effector proteins beta-catenin and MMP9.

## 1. Introduction

Even after treating cancer with current endocrine drugs and anticancer therapies, chemoresistance, relapse, and metastasis still present great challenges to curing this life-threatening disease. Obesity in cancer patients exacerbates this challenge to an even greater degree [1,2,3]. Despite the progress in understanding the association between adipose tissue biology and breast cancer, questions regarding how obesity-induced changes in white adipocytes regulate the complex dynamics of cell migration, loss of cell adhesion, invasion, and angiogenesis remain unanswered. Studies have confirmed that obese or dysfunctional white adipocytes stimulate breast cancer progression by increasing breast cancer cell survival, growth, proliferation, migration, and invasion [4,5,6,7,8]. However, which signaling pathways are exerting these biological effects remain unknown. Adipocyte-secreted factors may be critical in connecting obesity and the advanced progression of breast cancer via transducing signals from dysfunctional adipocytes in an autocrine, paracrine, and/or endocrine manner.

Extracellular signaling proteins stimulate changes in cell behavior through binding to different cell surface receptors and transducing signals through their intracellular secondary messenger proteins [9]. Focal adhesion kinase (FAK) is one such cytoplasmic secondary messenger protein that regulates multiple downstream cell signaling pathways controlling the biological phenomena of cell detachment, cell migration, invasion, and angiogenesis [10,11,12,13,14,15,16,17,18]. FAK is known to play an active role in controlling breast cancer cell migration, invasion, and angiogenesis by activating corresponding downstream effector proteins [19,20,21,22,23,24,25,26]. Studies have also shown how extracellular stimuli can activate FAK and relay downstream metastatic signals [27,28,29]. ECM ligands and other proteins, such as growth factors, cytokines, etc., control FAK activation and autophosphorylation [30]. Thus, being a mediator between extracellular stimuli and downstream effector proteins regulating metastatic events, we hypothesized that obesity-induced changes in the breast tumor microenvironment are likely to activate FAK in breast cancer cells. White adipose tissue (WAT), which forms the central portion of the breast tumor microenvironment, is recognized as an active endocrine organ secreting different bioactive compounds consisting of growth factors, hormones, cytokines, chemokines, adiponectin, and leptin [31]. However, due to excess energy intake, increased fat accumulated as triglycerides leads to the formation of dysfunctional adipocytes that increase in volume, but not in cell number [31]. This compromises the ability of the adipocytes to maintain their normal adipocytokine secretion patterns [32,33]. Furthermore, white adipocytes crosstalk in a paracrine manner with breast cancer cells through these adipocyte-secreted factors, which not only increases the risk of developing breast cancer due to hormone conversion, mediated through aromatase in post-menopausal woman, but also facilitates breast cancer cell migration and invasion away from the localized tumor [4,5,6,8,32,34,35]. However, studies have yet to find an association between obesity-induced changes in WAT and FAK activation followed by activation of downstream effector proteins regulating breast cancer metastasis.

The metastatic pathway activation, associated with obesity-mediated changes in white adipocyte secretions, is not fully understood. It certainly involves activation of diverse cell signaling pathways relating to metastasis. Along with executing the biological responses for cell survival and proliferation, FAK activation acts as a mediator of several events fine-tuning metastatic progression of cancer cells, including rearrangements of the cytoskeleton, promotion of epithelial to mesenchymal (EMT) traits, upregulation of MMPs to facilitate invasion, and extracellular matrix remodeling [36]. In short, FAK mediates several metastatic events through multifunctional activation of diverse cell signaling pathways in a kinase-dependent manner. In the current study, we investigated the expression of the key FAK downstream targets in this process: MMP-9 and β-catenin. The rationale for evaluating the expression of β-catenin in this study is based on the observations involving the crosstalk between FAK and the Wnt/β-catenin signaling pathway, where it is reported that FAK is required for β-catenin expression [37,38,39]. One study reported that FAK-mediated phosphorylation of GSK-3β inhibits destruction complex activity and frees more β-catenin for cancer-associated gene expression, thereby mimicking Wnt stimulation. Moreover, the tissue-specific expression of β-catenin protein in obesity-related cancer with respect to the non-obese condition is not well defined. It is reported that β-catenin expression can be controlled downstream by FAK activation [37], which has a role in cancer metastasis.

Beta-catenin is a central signal transducer of the canonical Wnt signaling pathway regulating diverse physiological processes from embryonic development to homeostasis, regeneration, and repair of adult tissue [40,41,42,43]. One important characterized function of this signaling pathway is the proliferation and self-renewal of stem cells [41,44]. Studies have reported abnormal expression of β-catenin protein in cancer cells, which, as a transcription factor, transactivates genes involved in cell division, stemness, and metastasis [45,46,47,48,49,50]. Generally, in the absence of a Wnt ligand, β-catenin is phosphorylated and ubiquitinated by the cytoplasmic destruction complex composed of APC, CK1, glycogen synthase kinase 3, and AXIN1 [45,46,51]. However, the presence of a Wnt ligand binding to the Frizzled receptor and low-density lipoprotein receptor protein 5/6 (LRP5/6) co-receptor induces a series of molecular events leading towards inactivation of the destruction complex and inhibition of β-catenin phosphorylation [46,51]. Thus, it freely accumulates and translocates to the nucleus to regulate target gene transcription by associating with the T cell factor/Lymphoid enhancer factor (Tcf/Lef) transcription factor [46,51].

Metastatic events in cancer cells are an intricate process involving their detachment through EMT, acquiring invasiveness by compromising the extracellular matrix (ECM), facilitating escape to body fluids. Under normal conditions, the ECM acts as a physiological barrier. However, due to the secretion of proteolytic enzymes from cancer cells, the degraded and compromised ECM becomes permissive for cancer cell invasion. An important family of proteolytic enzymes that degrade the ECM and basement membrane are matrix metalloproteinases (MMPs) [52,53]. Increased expression of MMP-9 has been reported in various cancer patients, and it is associated with their poor disease prognosis [54,55,56]. MMP-9 specifically degrades Type IV collagen, a major component of the basement membrane [57,58]. Thus, it removes the physical barrier through which metastatic cancer cells can make their way to circulation. Such reorganization of the ECM also facilitates the supply of necessary nutrients and oxygen to the growing tumor in situ by providing structural plasticity for angiogenesis [57]. MMP-9 also helps facilitate the process of EMT by cleaving and inactivating the cell junction protein E-cadherin, thus helping the cancer cell escape the primary tumor site [57,59,60]. The ability of MMP-9 to degrade the extracellular matrix makes it an important protein for the metastatic progression of cancer. 

The rationale for focusing on MMP-9 is based on studies that have also reported expression of MMP9 downstream to FAK activation or indirectly through the Src or the PI3K/Akt/NF-kB signaling axis [27,29,61,62,63,64,65,66]. Studies have found increased expression of MMP-9 protein with respect to the presence of several endocrine factors that are secreted from adipocytes [27,29,41,61,64,65,67]. The binding of those endocrine factors to this receptor tyrosine kinase protein induces activation of a downstream signaling cascade involving the activation of the transcription factor NF-kB. The NF-kB signaling pathway plays a major role in MMP-9 expression and secretion by binding to the specific sequences of the MMP9 gene promoter and triggering its transcription [59,68]. Additionally, the connection of Wnt signaling and MMP-9 expression has been extensively studied in various cancers. Elevated expression of Wnt3A/β-catenin and MMP-9 have been reported in primary tissue of colorectal cancer through immunohistochemical staining [69]. Another study reported increased β-catenin expression in nasopharyngeal carcinoma and that β-catenin inhibition through siRNA transfection suppresses MMP-9 expression as a downstream target [70]. Based on these observations, the current study addresses whether the aberrant changes in adipokines under conditions mimicking obesity can increase the post-translation modification of FAK protein through phosphorylation at the Tyr397 position in breast cells. Thus, we hypothesized that under conditions mimicking obesity, an increase in β-catenin accumulation and MMP-9 protein expression would occur downstream of FAK activation, leading to a subsequent increase in breast cancer cell migration and invasion. To elucidate the association of this signaling pathway downstream to cell surface receptors activated by adipocyte-secreted factors at the molecular level, this study focused on the activation of FAK protein through its autophosphorylation at the Tyr397 position in breast cells.

To address this hypothesis and explore the role of activated FAK in potentiating breast cancer metastasis, we utilized an obese adipocyte differentiation model describing the process in 3T3-L1 MBX cells [71]. In this previous study, we observed and reported increased differentiation yields as well as lipid accumulation and adipokine status when their differentiation was maintained in the presence of a fatty acids cocktail [71]. Our analysis of the secretion profiles of 3T3-L1 MBX adipocyte cells, when they were exposed to the fatty acids cocktail, revealed that there were changes in the release of key adipokines in comparison to the control differentiated cells with no fatty acids and yielded a molecular overview of the successful differentiation and maintenance of adipogenesis characteristics in 3T3-L1 MBX adipocytes. PPAR-γ, a master regulator of terminal adipogenesis, plays a vital role in adipocyte biology by transcriptionally regulating the physiological process from their development to the maintenance of metabolism [72,73,74]. For review of the interactions of obesity-related adipokines and breast cancer metastasis, please see Mubtasim et al. [75].

## 2. Results

### 2.1. Exposure to the In Vitro Fat Cell-Derived Conditioned Medium Mimicking Obesity Signaling Increased the Migratory Potential of Breast Cells

To investigate the effect of adipocyte-secreted factors on the migratory potential of breast cells, a wound healing assay using a novel Wound Assay Device [76] was performed. Images at time 0 represent the area of the wound at the time of scratch formation. Images in Figure 1 show the cells that migrated to the wound area at time 24, 48, and 72 h. Breast cancer cells MDA-MB-231 and MCF7 took ~72 h and 96 h, respectively, to close the wound completely for many of the treatment conditions. Interestingly, non-tumorigenic breast cells MCF-10A were faster in closing the wound with exposure to the adipocyte-derived conditioned medium in comparison to MCF7. MCF-10A cells also took ~72 h to close the wound completely for many of the treatment conditions. Both MCF7 (Figure 1(Aa,Ba)) and MDA-MB-231 (Figure 1(Ab,Bb)) cell lines, exposed to in vitro mature and obese adipocyte-derived conditioned media, were faster in closing the wound than the untreated cells. Highly metastatic MDA-MB-231 breast cancer cells showed enhanced migratory capabilities in closing ~80% of the wound in 48 h when cultured with obese adipocyte-derived conditioned medium than the low metastatic MCF7 breast cancer cell after 72 h. For MCF7 cells, culture with pre- and mature adipocyte-derived conditioned media was also very successful in promoting wound closure to 80% in 72 h versus untreated MCF7 cells. Interestingly for the non-tumorigenic MCF-10A cells, preadipocyte-derived conditioned medium was the most effective in closing 80% of the wound in 48 h (Figure 1(Ac,Bc)) versus the other treatments (mature- and obese-derived adipocyte conditioned media). Moreover, exposure to pre-adipocyte-derived conditioned medium promoted MCF-10A cells to close the wound faster than the MCF-10A cell exposed to their standard culture medium supplemented with 10% FBS (Positive Control) as mentioned in the Methods.

### 2.2. Exposure to the In Vitro Fat Cell-Derived Conditioned Medium Mimicking Obesity Increased the Invasive Potential of Breast Cells

To investigate the effect of adipocyte-secreted factors on the invasive potential of breast cells, a Matrigel™ invasion assay using Transwell™ inserts was performed. Non-tumorigenic (MCF-10A), low metastatic (MCF7), and high metastatic (MDA-MB-231) were treated with in vitro fat cell-derived conditioned medium for 72 h. As expected, results for the untreated MCF7 (Figure 2D) in comparison to untreated MDA-MB-231 (Figure 2E) showed that MDA-MB-231 cells have higher invasive potential than MCF7 cells. The data also showed that exposure to in vitro obese adipocyte-derived conditioned medium caused a higher number of MCF7 cells (Figure 2D) to invade through the Matrigel™ in comparison to the untreated, preadipocytes and mature adipocyte-derived conditioned media. The MDA-MB-231 cells (Figure 2E) showed similar results. Non-tumorigenic MCF-10A (Figure 2F) cells showed significantly less invasion potential than the untreated MCF7 and MDA-MB-231 cells. Moreover, their exposure to in vitro fat cell-derived conditioned medium made them less invasive than untreated MCF-10A cells (Figure 2F). Interestingly, when MCF-10A cells were exposed to preadipocyte-derived conditioned medium, their invasiveness increased to that of the untreated MCF-10A.

### 2.3. Exposure to the In Vitro Fat Cell-Derived Conditioned Medium Mimicking Obesity Activates FAK Protein through Its Phosphorylation at Tyr 397 Position in Breast Cancer Cells

Extracellular signaling molecules such as growth factors, cytokines, etc., control FAK activation through autophosphorylation at Tyr397 [30]. With the changes in the secretion pattern of adipocytes, we investigated whether there were any changes in the post-translation modification status of FAK protein through its autophosphorylation at Tyr397. Figure 3A shows representative immunoblotting results of phosphorylated p-FAK (Tyr397), total FAK, α-tubulin, and β3 tubulin protein levels in breast cells, exposed to in vitro fat cell-derived conditioned media. When MCF7 cells were exposed to in vitro fat cell-derived adipocyte-secreted factors, an upward trend of autophosphorylation of FAK protein was observed under obese conditions (Figure 3B). With the same exposure conditions, MDA-MB-231 cells showed a significant increase in the fold change expression of p-FAK protein at Tyr-397 with respect to the untreated MDA-MB-231 (Figure 3C). For non-tumorigenic MCF-10A cells, no change in the phosphorylation status of FAK was observed when they were exposed to in vitro mature and obese adipocyte-derived conditioned medium (Figure 3D). However, for the pre-adipocyte-derived conditioned medium, MCF-10A cells showed an upward trend of phosphorylation status of FAK protein in comparison to the untreated (Figure 3D).

### 2.4. Exposure to the In Vitro Fat Cell-Derived Conditioned Medium Increases β-Catenin Stabilization in Breast Cells

Aberrant expression of beta-catenin, an essential intermediate in the Wnt signaling pathway, has been reported for many malignancies [77,78]. β-catenin protein was found to be stabilized downstream to the FAK protein-activated cell signaling pathway [37,39]. Hence, in this study, we investigated whether, with the changes in the secretion pattern of adipocytes, there were any changes in the stabilization for β-catenin protein. For that purpose, we exposed MCF-7, MDA-MB-231 and MCF-10A cells to the varied phases of adipocyte-derived conditioned media for 2 and 8 h. We then analyzed the changes in expression of beta-catenin in the whole cell lysate of MCF-7, MDA-MB-231, and MCF-10A with respect to their treatment by Western blotting. Figure 4A is a representative blotting image of beta-catenin and α-tubulin protein expressed in breast cells exposed to in vitro fat cell-derived conditioned medium. The data showed that MCF7 cells (Figure 4B), exposed to in vitro adipocyte-derived conditioned medium mimicking obesity, had significant elevation in β-catenin stabilization after 2 h, compared to untreated. β-catenin stabilization continued to remain elevated over the 8 h exposure of in vitro adipocyte-derived conditioned medium mimicking obesity. However, this intensity was much less than the expression found at 2 h of exposure under the same conditions. For MDA-MB-231 cells (Figure 4C), the elevation in β-catenin stabilization in cells exposed to in vitro adipocyte-derived conditioned medium mimicking obesity is not significant, but the trend is upward, both for short and long exposure times. Importantly, no changes in the stabilization of β-catenin were observed for non-tumorigenic MCF-10A cells (Figure 4D) exposed to any of the in vitro adipocyte-derived conditioned media irrespective of adipogenic changes or conditions mimicking obesity.

### 2.5. Exposure to the In Vitro Fat Cell-Derived Conditioned Medium Increases MMP-9 Protein Expression in Breast Cancer Cells

Studies have shown increased expression of MMP-9 protein with respect to the presence of several endocrine factors [27,29,41,61,64,65,67]. Hence, in this study, we investigated whether the changes in the secretion pattern of adipocytes (specifically under conditions mimicking obesity) caused changes in the expression of MMP-9. To address this, MCF7, MDA-MB-231, and MCF-10A were exposed to various adipocyte-derived conditioned media for 2 and 8 h. Following this, changes in expression of MMP-9 in the whole cell lysate of MCF7, MDA-MB-231, and MCF-10A with respect to their exposure conditions of adipocytes were analyzed by Western blotting. Figure 5A shows a representative blotting image of MMP-9 and α-tubulin protein expressed in breast cells exposed to in vitro fat cell-derived conditioned medium. The data indicate constant, significant elevation of MMP9 expression in MCF7 (Figure 5B) and MDA-MDB-231 (Figure 5C) cells both for 2 and 8 h exposure to the adipocyte-secreted factors mimicking obesity, in comparison to untreated. The elevation in MMP-9 expression was also significantly elevated in MDA-MB-231 cells exposed to in vitro mature adipocyte-derived condition medium for 8 h. However, no changes in the expression of MMP9 were observed for non-tumorigenic MCF-10A cells (Figure 5D) exposed to any of the in vitro adipocyte-derived conditioned media, irrespective of adipogenic changes or conditions mimicking obesity.

## 3. Discussion

Obesity has long been considered a trigger for the metastatic progression of cancer [79]. Obesity exerts its molecular effects through white adipose tissue and its secreted endocrine factors [32,33,80]. For a critical review of the interactions between obesity-associated changes in white adipocytes and breast cancer progression as well as the current findings, see Mubtasim, et al. [75]. The purpose of the current study was to determine the effect of adipocytokine secretions from adipocytes as an environmental instigator for the metastatic progression of breast cancer. This study reported how the adipocytokine secretion from the in vitro 3T3-L1 MBX cell-derived adipocytes, in their different adipogenic forms, preadipocyte, mature adipocyte, and obese adipocyte (adipocyte under condition mimicking obesity) [71], plays a role in shaping the behavior of breast cells with respect to migration and invasion. Both low (MCF-7) and high metastatic (MDA-MB-231) breast cancer cells were found to be significantly affected by the adipocytokine secretions from the obese adipocytes. In their presence, breast cancer cell motility and invasiveness were increased in comparison to untreated MCF-7 and MDA-MB-231 cells. Secretions from pre- and mature adipocytes also enhanced the motility of MCF-7 breast cancer cells. However, the effect of those similar exposures on the invasiveness of MCF-7 breast cancer cells was not found to be significant. Likewise, secretions from mature adipocytes also elevated the migratory potential of MDA-MB-231 breast cancer cells without significant effects on their invasiveness. The motility and invasiveness of non-tumorigenic breast cells MCF-10A were not significantly affected by exposure to in vitro mature or obese adipocytes. However, and interestingly, their motile and invasive behavior were significantly impacted by adipocyte secretions from preadipocytes in comparison to untreated MCF-10As. This unexpected finding of preadipocyte secretions on non-tumorigenic cells needs further exploration.

The possible involvement of the FAK-mediated cell signaling pathway by which adipocytokine secretions may drive the migration and invasion of breast cancer cells was addressed in previous studies [27,28,29,30]. Studies have also reported an associated increase in FAK protein activation through its phosphorylation with breast cancer progression [12,13,14,15,16,17,18]. In this current study, we report, for the first time, that adipocytokines secreted in vitro from obese adipocytes [71] significantly increased FAK activation (through phosphorylation) in both low (MCF7) and high (MDA-MB-231) metastatic breast cancer cells. The increased FAK autophosphorylation of MCF-7 and MDA-MB-231 cells exposed to obese adipocyte-derived conditioned medium is correlated with the increased migratory and invasive potential of MCF-7 and MDA-MB-231 cells reported in this current study. This observation could possibly be due to overexpression of specific receptors in breast cancer cells that become activated in the presence of aberrant secretion of adipocytokines from obese adipocytes. Confirmation of this observation needs further investigation. In contrast, and interestingly, the phosphorylation status of FAK in non-tumorigenic breast cells MCF-10A were not significantly influenced by adipocytokine secretions from in vitro adipocyte-derived conditioned media. 

FAK plays an active role in controlling breast cancer cell migration, invasion, and angiogenesis by activating corresponding downstream effector proteins [19,20,21,22,23,24,25,26]. The current study also investigated the effect of adipocytokines towards influencing the expression of the key prognostic marker proteins for cancer progression, β-catenin and MMP-9. The results showed that adipocytokines from in vitro adipocytes mimicking obesity [71] increased the stabilization of β-catenin and the expression of MMP-9 protein in MCF-7 and MDA-MB-231 breast cancer cells, in comparison to pre and mature adipocytes, or in untreated conditions. Increase in MMP-9 expression in MCF-7 and MDA-MB-231 breast cancer cells exposed to adipocytokines from in vitro differentiated obese adipocytes is correlated with the increase in the development of invasiveness observed in the cells in the current study. This suggests that obesity impacts breast cancer progression by increasing the expression of key marker proteins of metastasis (i.e., MMP-9 and β-catenin) through FAK activation. Interestingly, comparing these results with the non-tumorigenic MCF-10A cells, no such changes were observed in the expression of either β-catenin or MMP-9 when exposed to adipocytokine secretions from any form of adipocytes. Another important finding for non-tumorigenic MCF-10A cells was that its FAK phosphorylation, MMP-9 expression, and β-catenin stabilization were not influenced by the preadipocyte-derived conditioned medium. These findings suggest that preadipocyte secretions do not influence the migratory or invasive properties of MCF-10A cells observed following FAK activation or by influencing β-catenin or MMP-9 expression.

The current study results present a molecular overview of the broader picture of how obesity, as an environmental instigator, may impact cancer progression in obese breast cancer patients [81] through a proposed cell signaling pathway mediated by FAK phosphorylation (Figure 6). An in vitro fat cell model mimicking obesity [71] has an impact on the migratory and invasive behavior of breast cancer cells. These [71] results confirm experimental reports from previous studies [4,5,6,32,34,35]. This is the first study to report that fat cells developed in vitro [71], did not have any impact on the migratory and invasive behavior of nontumorigenic breast cells. Additionally, this study shows that phosphorylation of FAK in breast cancer cells is increased upon exposure to adipocyte secretions collected from fat cells developed in vitro under conditions mimicking obesity [71]. However, no such changes in the phosphorylation of FAK were observed in non-tumorigenic breast cells. Furthermore, this study has provided molecular evidence for increased beta-catenin accumulation and MMP-9 expression in breast cancer cells upon exposure to adipocyte secretions collected from in vitro-derived obese adipocytes [71]. No such effect was observed for non-tumorigenic breast cells. (Figure 6). Taken together, these molecular observations suggest that adipocyte secretions, under conditions mimicking obesity, can promote the metastatic behavior in breast cancer cells towards a more aggressive phenotype without impacting normal breast cells. However, this study does have some limitations. The study has evaluated the effect of adipocytokines through a conditioned medium approach. It has evaluated the effect of murine-derived adipocytes on human breast cancer cells. MMP-9 is a secretory protein, and the current study observed the changes in MMP-9 expression in the whole cell lysate of breast cancer cells using Western blotting. This may explain why more robust and significant changes in protein expression were not observed in the current study.

In summary, this study showed adipocyte secretions under conditions mimicking obesity can instigate cancer progression, activating the FAK cell signaling pathway, stabilizing beta-catenin protein, and increasing MMP-9 expression. Future work can include exploring the expression or post-translational modification of proteins immediately downstream to auto phosphorylated FAK to further validate the activation of the FAK signaling pathway. Furthermore, FAK knockout breast cell models could be developed to establish the connection between β-catenin and MMP9 expression downstream to FAK activation. Studies to establish MMP-9 expression, downstream of FAK protein activation and β-catenin stabilization (by performing FAK knockout and luciferase promoter reporter chromatin immunoprecipitation assays) in breast cancer cells, would also elucidate these relationships. Adipocyte and breast cells or breast cancer cells mainly crosstalk in the microenvironment through paracrine factors secretion. The current study has only shown crosstalk in one direction—from adipocytes to breast cells. However, the observations may differ if bi-directional crosstalk was included. The bi-directional crosstalk events between adipocytes and breast cells could also be better understood by designing experiments with co-cultured cells from either in vitro or in vivo origins.

## 4. Materials and Methods

To address the potential relationship of adipocyte-secreted factors towards activating FAK and its downstream effector proteins, non-tumorigenic breast cells (MCF-10A) and breast cancer cells with low (MCF7) and high (MDA-MB-231) metastatic potential were exposed to in vitro murine adipocyte-derived, conditioned medium (CM). For the in vitro adipocyte model, 3T3-L1 MBX cells differentiated to obesity with fatty acids supplementation were used [71]. Murine 3T3-L1 MBX fibroblasts exhibited a pre-adipocyte form (fibroblast), a differentiated form (mature adipocytes), and with addition of fatty acid supplementation, an obese form [71]. These forms were used to create separate control in in vitro conditioned medium treatment environments [71]. The conditioned medium containing adipocyte-secreted factors were explored using a proteome profiler mouse adipokine array kit as previously described [71]. The media containing secreted factors from these different forms of adipocytes were used as conditioned media derived from in vitro fat cells in this current study [71].

### 4.1. Cell Culture

The breast cancer cell lines MCF7 and MDA-MB-231 were cultured in high glucose Dulbecco’s Modified Eagle Medium (DMEM; Gibco, Catalogue# 12800-017, Waltham, MA, USA) supplemented with 10% Fetal Bovine Serum (FBS; ATLAS Biologicals, Catalogue# F-0500-D, Fort Collins, CO, USA) and 1% Penicillin and Streptomycin (Thermo Fisher Scientific, Catalogue # 15140-122; Waltham, MA, USA) in a humidified atmosphere of 5% CO_2_ at 37 °C. The nontumorigenic cell line MCF-10A was cultured in DMEM/Ham’s F-12 (Invitrogen, Catalogue # 11330-032; Carlsbad, CA, USA) supplemented with 5% horse serum (Invitrogen, Catalogue# 16050-122; Carlsbad, CA, USA), 20 ng/mL recombinant human epidermal growth factor (EGF; Catalogue# PHG0311Gibco, Waltham, MA, USA), 0.5 µg/mL hydrocortisone (Sigma-Aldrich, Saint Louis, MO, USA), 100 ng/mL cholera toxin (Sigma-Aldrich, Catalogue# C8052, Saint Louis, MO, USA), 10 mg/mL insulin (Sigma-Aldrich; Catalog # I0516, Saint Louis, MO, USA), and 1% Penicillin/Streptomycin (Gibco, Catalogue # 15140-122, Waltham, MA, USA) in a humidified atmosphere of 5% CO_2_ at 37 °C.

### 4.2. Wound Healing Assay

To determine changes in migration potential, a novel Wound Assay Device was employed. Initially, 200 k–250 k cells based on the cell line phenotype were plated onto each well of 12-well plates for 90–100% confluency in 24 h. On the day of wound creation, a novel Wound Assay Device [76], invented to specifically produce accurate, reproducible wounds/gaps, was used to form the gap across the well. Following this, the media and any cell debris were aspirated carefully from the wells with PBS. This PBS was discarded, and DMEM media containing 0.5% FBS, conditioned with appropriate treatments, were added slowly against the well wall to avoid detaching additional cells from each well. Following the generation and inspection of the wound, an initial image focusing on the gap generated on each well was taken using an EVOS XL Core imaging system (Thermo Fisher Scientific; Catalog# AMEX1000; Waltham, MA, USA) to measure the area of the gap generated at time 0 h. After imaging, 12-well plates were placed incubated at 37 °C and 5% CO_2_). At 24, 48 and 72 h, images were taken of each well plate, focusing on the generated gap, to measure the closure of the gap against time.

### 4.3. Transwell™ Matrigel™ Invasion Assay

BD Matrigel matrix (BD Biosciences, Catalogue# 356234; Franklin Lakes, NJ, USA) aliquots were thawed on ice at 4 °C overnight. Once thawed, vials were agitated to ensure that contents were evenly dispersed. The vials were kept on ice throughout and handled under aseptic conditions. Pipets, syringes, or containers that came into contact with the Matrigel™ matrix were chilled prior to use. To dilute the Matrigel™ matrix (8 mg/mL) solution, it was mixed in ice cold FBS-free DMEM to a final concentration of 250 µg/mL. The lid from a 24-well Transwell™ plate was removed and using a chilled sterile pipet, and each Transwell™ insert was coated with 80 µL of the diluted Matrigel™ matrix solution. The coated Transwell™ plates were then incubated at 37 °C for 2 h. Following this, the breast cells were detached from the tissue culture plate using a 0.25% Trypsin-EDTA solution (Thermo Fisher Scientific, Catalog# 25200056; Waltham, MA, USA) and pelleted by centrifugation, after which the supernatant was aspirated, leaving behind the pelleted cells. These pelleted cells were resuspended in serum-free DMEM media. Around 100 k cells (suspended in <100 µL) were added into the upper compartment of the insert and incubated for 10 min at 37 °C and 5% CO_2_ to allow the cells to settle down. Using a pipette, 700 μL of the DMEM was very carefully added for each treatment condition into the lower chamber of the Transwell™ inserts in such a way that the chemoattractant liquid or treatments in the bottom well make contact with the membrane of the inserts to form a chemotactic gradient. Following this critical step, the Transwell™ plates were incubated for 72 h under normal culture conditions. On the day of analysis, the medium inside the insert was carefully aspirated. About 600–1000 μL of 0.2% crystal violet was added into each well and the inserts were repositioned into it for staining, followed by incubation at room temperature for 5–10 min. The Transwell™ inserts were washed gently with distilled water to remove unbound crystal violet and then air-dried. The cells on the inside of the Transwell™ were gently removed using cotton swabs. To measure the cells that had migrated through the Transwell™pores, an acetic acid elution of 33% (*v*/*v*) with ddH_2_O was used. The bound crystal was eluted by adding 400 µL of 33% acetic acid into each insert. It was then gently shaken for 10 min. The eluent from the lower chamber was then transferred to a 96-well, clear microplate, and the absorbance was measured at 590 nm using a Bio-Tek Synergy H1 Microplate Reader H1M (Bio-Tek; Winooski, VT, USA).

### 4.4. Protein Extraction and Quantification

Lysis buffer for protein extraction was prepared by mixing Pierce RIPA lysis buffer (Lot# XG348655, Thermo Scientific) with Halt Protease, Phosphatase inhibitor (100×), and 0.5 M EDTA (catalog# 78420, 78420, 87886; Thermo Fisher Scientific, Waltham, MA, USA) Cocktail. The inhibitor cocktail contains a broad spectrum of protease inhibitors (AEBSF, aprotinin, E-64, bestatin, leupeptin, and pepstatin A) and phosphatase inhibitors (sodium fluoride, sodium orthovanadate, sodium pyrophosphate, and beta-glycerophosphate). For every 500 µL of lysis buffer, 5 µL of 100× protease, phosphatase, and EDTA was added to a final concentration of 1×. To prepare the whole-cell lysate of protein extract, the existing medium was discarded, and the cells were washed in cold FBS-free DMEM media. This was discarded, and 500 µL of ice-cold lysis buffer was added. The culture dish was put at −80 °C for 3 min. Afterwards, the cells were scraped using an ice-cold plastic cell scrapper and collected in a microcentrifuge tube. The contents in the centrifuge tube were agitated for 15 min and then were centrifuged at 12,000 rpm for 15 min at 4 °C. After centrifugation, the cell pellet was discarded, and the cell supernatant was collected as a whole cell lysate of protein extract. The BCA protein assay was used to determine the protein concentration of the whole cell lysate of MCF-7, MDA-MB-231, and MCF-10A for different treatment conditions at different time points. Reference standard protein (albumin) concentrations were made as per the manufacturer’s instructions (Thermo Scientific). A total of 200 µL of protein analysis solution was added to 25 µL of standard and sample proteins, in duplicates, as per the instructions. Finally, reference standard and sample protein concentrations were quantified with the 562 nm laser on a Bio-Tek Synergy H1 Microplate Reader H1M.

### 4.5. SDS-Polyacrylamide Gel Electrophoresis and Immunoblotting

An amount of 40 µg of protein from the whole cell lysate of MCF7, MCF-10A, and MDA-MB-231 was mixed with 6X Laemmli sample in a 5:1 ratio. The remaining volume was adjusted using RIPA lysis buffer. The prepared loading samples were denatured by heating at 95 °C for 5 min. Samples were then loaded to each well of a hand-cast SDS-polyacrylamide gel. This hand-cast polyacrylamide gel was composed of 8–10% gradient resolving gel and 4% stacking gel. For high molecular weight proteins (MW > 250 kDa), a 6–10% gradient resolving gel and 4% stacking gel were used. EzRun Prestained Rec Protein Ladder (Fisher Bioreagents, Lot# 0015688; Pittsburgh, PA, USA) was used as size reference. The denatured lysates were resolved on the gradient polyacrylamide gel at 120 V until the 17 kDa protein reached the bottom of the gel. A wet transfer technique was used to transfer the protein from the gradient polyacrylamide gel to the nitrocellulose membrane (Thermo Scientific, Lot # XE3443891). The wet transfer was done in an ice bath with 1 × transfer buffer (25 mM Tris; 192 mM glycine) at 110 V for 60 min. After transfer, Ponceau S solution (Thermo Scientific, Catalogue# A40000279) was used to assess efficiency of protein transfer. Nitrocellulose membranes were blocked with 3% bovine serum albumin blocking buffer made in 1× TBS-Tween buffer (Trizma HCl, NaCl, ultra-pure water, Tween 20) overnight. Following blocking, membranes were incubated with primary antibodies: rabbit phospho-FAK monoclonal antibody at the Tyr397 position (Cell Signaling Technology, Catalogue# 3283S; Danvers, MA, USA), mouse FAK monoclonal antibody (Proteintech, Catalogue# 66258-1-Ig; Rosemont, IL, USA), mouse β-catenin monoclonal antibody (Proteintech, Catalogue# 66379-1-Ig; Rosemont, IL, USA), rabbit MMP-9 polyclonal antibody (Proteintech, Catalogue# 10375-2-AP; Rosemont, IL, USA), mouse alpha tubulin monoclonal antibody (GenScript, Catalogue# A014010; Piscataway, NJ, USA), and mouse β3-tubulin monoclonal antibody (Cell Signaling Technology, Catalogue# 4466S) at 1:1000 dilution for 2 h. After the primary antibody incubation, membranes were washed with 1× TBS-Tween buffer 3 times for 5 min each and then incubated with Goat Anti-Mouse IgG HRP conjugated (Lot# A1014; Santa Cruz Biotechnology, Dallas, TX, USA) and Goat Anti Rabbit IgG HRP conjugated (Product # 7074, Cell Signaling Technology) secondary antibody for 90 min. Membranes were further washed prior to ECL (Thermo Scientific, Catalogue # 34577) exposure and visualized using the LiCor Odessey Imaging system (Odyssey^®^ Fc) by LI-COR Biosciences (Lincoln, NE, USA). The relative protein densities of the same protein from different samples of 3T3-L1 MBX cells were analyzed using Image J software(Version 10.0.2). The developed membrane was stripped using Restore PLUS Western Blot Stripping Buffer (Thermo Scientific, Lot# UJ291024). The stripped membrane was blocked, washed in between the primary and secondary antibody incubation, and developed for another protein, repeating the experimental steps described above.

### 4.6. Statistical Analysis

The statistical interpretation of the data was performed using GraphPad Prism 9 software (GraphPad Software Inc., San Diego, CA, USA). Each experiment was conducted independently 3 times with replicates, and the resultant data were expressed as mean ± standard error. The statistical significance between the treatment group and control was determined using analysis of variance (ANOVA). Based on the independent variables, one-way or two-way ANOVA were performed. Tukey’s post hoc test was employed for multiple comparison between group means. In some samples of wound healing data, Sidak’s multiple comparison was employed to counteract the possibility of family-wise error rate. The family-wise error rate is the chance of deriving at least one false conclusion or type I error in a series of hypothesis tests. The chance for family wise error rate increases for studies that make multiple comparisons. To counteract that possibility, in some of the wound healing assay samples, GraphPad Prism suggested using the Sidak multiple comparison test for this study. Overall, *p* value < 0.05 was considered statistically significant.

## 5. Conclusions

The current study has shown that adipocyte secretions under conditions mimicking obesity can impact breast cancer cell migration and invasion. The study has also generated important information on how adipocytokines from obese adipocytes can activate the FAK-mediated cell signaling pathway, triggering metastatic progression in cancer cells. Our results also show that β-catenin protein stabilization and MMP-9 protein expression become elevated with exposure to adipocytokines from obese adipocytes. Importantly and interestingly, adipocytokines from obese adipocytes do not exert influence on non-tumorigenic cells through these FAK-mediated factors towards invasion.

## Figures and Tables

**Figure 1 ijms-24-16605-f001:**
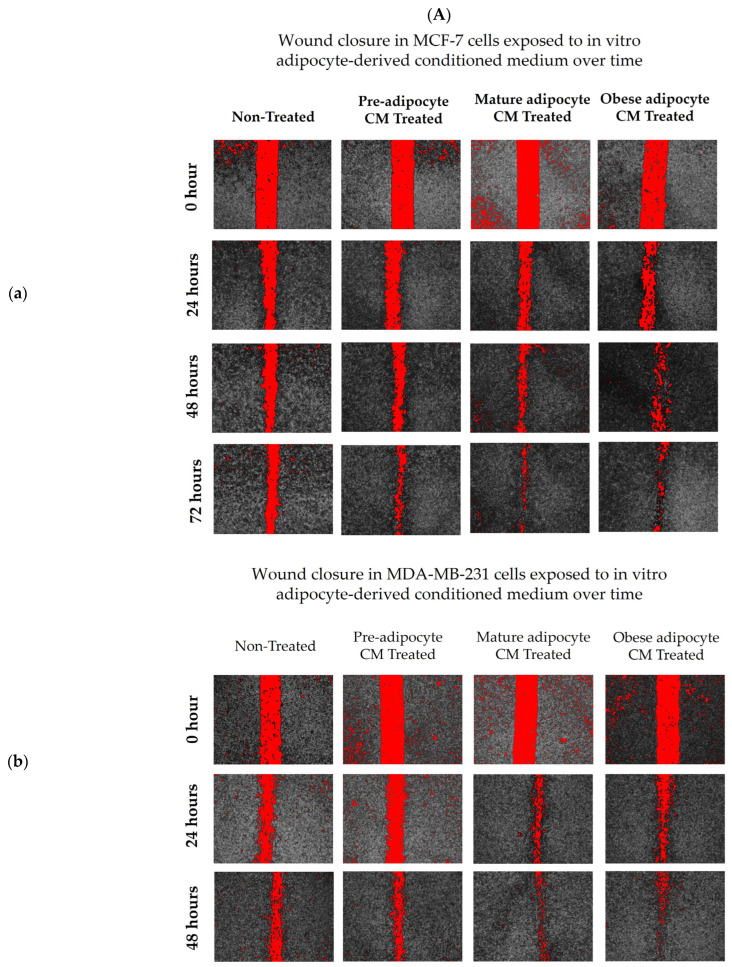
Adipocyte-secreted factors, mimicking obesity, influence the migratory potential of breast cells. (**A**) Representative photomicrographs. (**B**) Shows quantitation of wound closure in a cell monolayer of MCF7 (**a**), MDA-MB-231 (**b**), and MCF-10A (**c**) cells exposed to in vitro adipocyte-derived conditioned medium over time. The bar graphs (**B**) represent the percentage of wound closure for each treatment condition from time 0 as Mean ± SEM derived from three technical replicates of three independent experiments (in vitro). Positive control—MCF-10A cell in parent medium for 48 h. Untreated—MCF-10A cells treated with DMEM formulation used for MCF7 and 231 cells. The percentage of wound closure at each time point (24, 48 and 72 h) was calculated with respect to the area of wound created at time 0 corresponding to each condition. The significance for percentage of wound closure, among the treated conditions compared to untreated, over time for each cell line was measured using a Mixed Effect Model followed by Tukey’s post hoc test for multiple comparison. In some cases, Sidak’s multiple comparison was employed to counteract familywise error rate. *p*-value ≤ 0.05 was considered statistically significant. One asterisk (*) indicates *p* ≤ 0.05, two asterisks (**) indicate *p* ≤ 0.01, three asterisks (***) indicate *p* ≤ 0.001, and four asterisks (****) indicate *p* ≤ 0.0001. *p*-value > 0.05 was considered statistically nonsignificant (ns) and not marked on the graph to avoid clutter.

**Figure 2 ijms-24-16605-f002:**
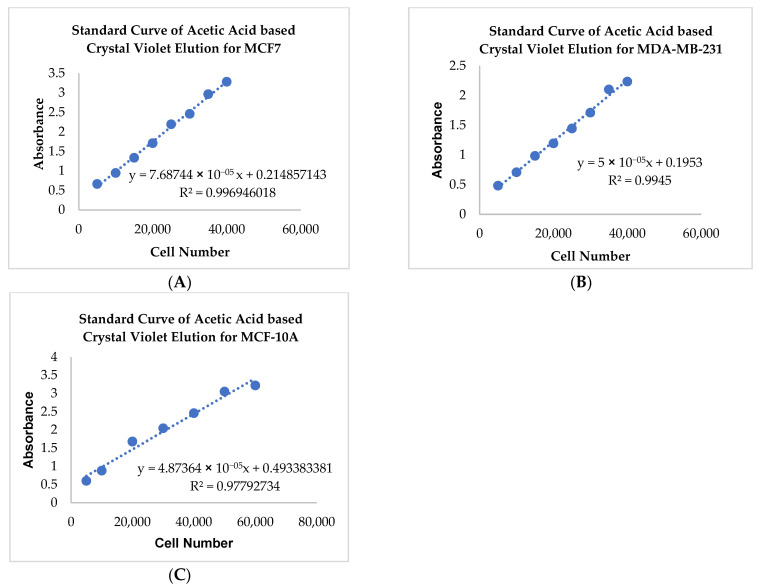
Adipocyte-secreted factors under conditions mimicking obesity influence the invasive potential of breast cells. The standard curve represents the acetic acid-based elution of crystal violet, stained across various cell densities of MCF7 (**A**), MDA-MB-231 (**B**), and MCF-10A (**C**) plated on a 96-well plate. The bar graph represents the number of MCF7 (**D**), MDA-MB-231 (**E**), and MCF-10A (**F**) cells that invaded on exposure to in vitro fat cell-derived conditioned media at 72 h. The number of invading cells, represented as Mean ± SEM, was determined by at least two technical replicates of three independent experiments (in vitro). The significance for the number of invading cells among the treated conditions with respect to the untreated, over time for each cell line, was measured using one-way ANOVA followed by Tukey’s post hoc Test for multiple comparison. A *p* value ≤ 0.05 was considered statistically significant. One asterisk (*) indicates *p* ≤ 0.05, and two asterisks (**) indicate *p* ≤ 0.01. *p*-value > 0.05 was considered statistically nonsignificant (ns) and not shown on the graph to avoid clutter.

**Figure 3 ijms-24-16605-f003:**
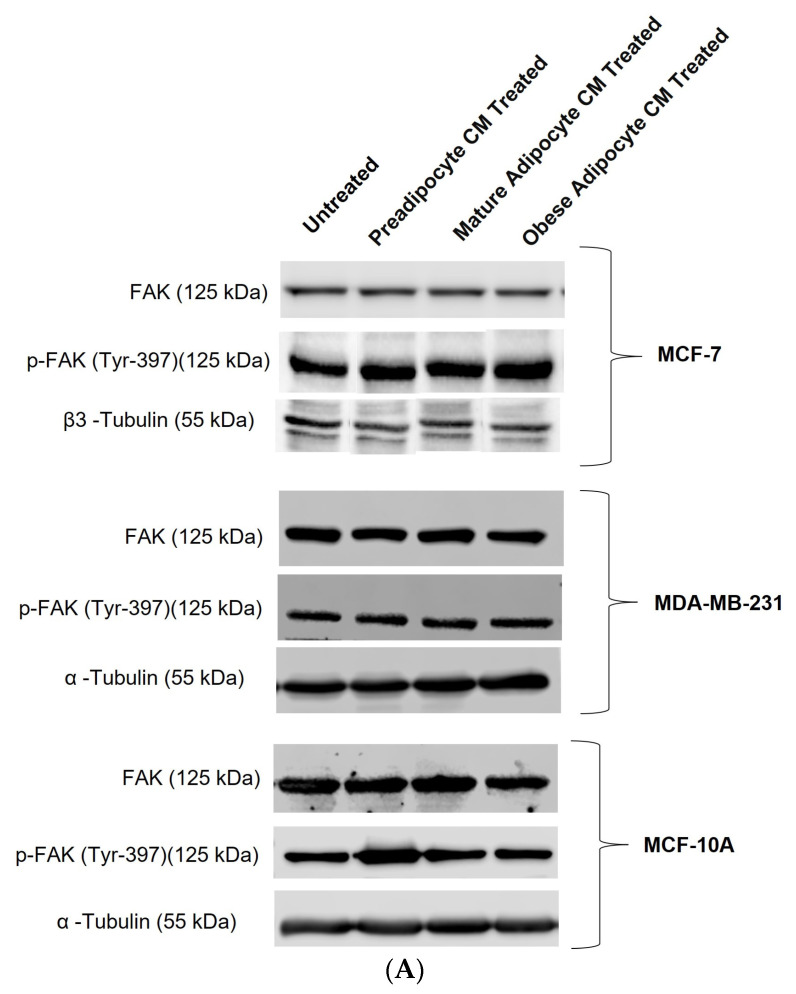
Adipocyte-secreted factors under obese conditions mimicking obesity elevates phosphorylation of FAK protein in breast cancer cells. MCF7, MDA-MB-231, and MCF-10A were exposed to in vitro fat cell-derived conditioned media treatments for 50 min. Immunoblotting on total lysates of each cell type, under each treatment condition, was performed using specific antibodies. (**A**) Representative Western blot showing phosphorylated p-FAK (Tyr397), total FAK, and loading control α-tubulin and β3 tubulin protein levels. The signal for p-FAK and FAK were normalized with densitometric analysis with respect to the loading control (α-tubulin and β3 tubulin signals) corresponding to each treatment. The bar graph represents fold change expression of p-FAK in mean ± SEM for MCF7 (**B**), MDA-MB-231 (**C**), and MCF-10A (**D**) when exposed to in vitro fat cell-derived conditioned media treatments, respectively. The fold change expression was derived by dividing the normalized expression of p-FAK protein for each treated condition with respect to the normalized expression of p-FAK protein for untreated conditions for each cell line. The mean value of the fold change expression of p-FAK protein for each condition was derived from at least 3 independent experiments. The significance for the changes in p-FAK expression among the treated conditions, compared to untreated, for each cell line was measured using one-way ANOVA followed by Tukey’s post hoc Test for multiple comparison. *p* value < 0.05 was considered statistically significant. One asterisk (*) indicates *p* ≤ 0.05. *p*-value > 0.05 was considered statistically nonsignificant (ns) and not shown on the graphs to avoid clutter.

**Figure 4 ijms-24-16605-f004:**
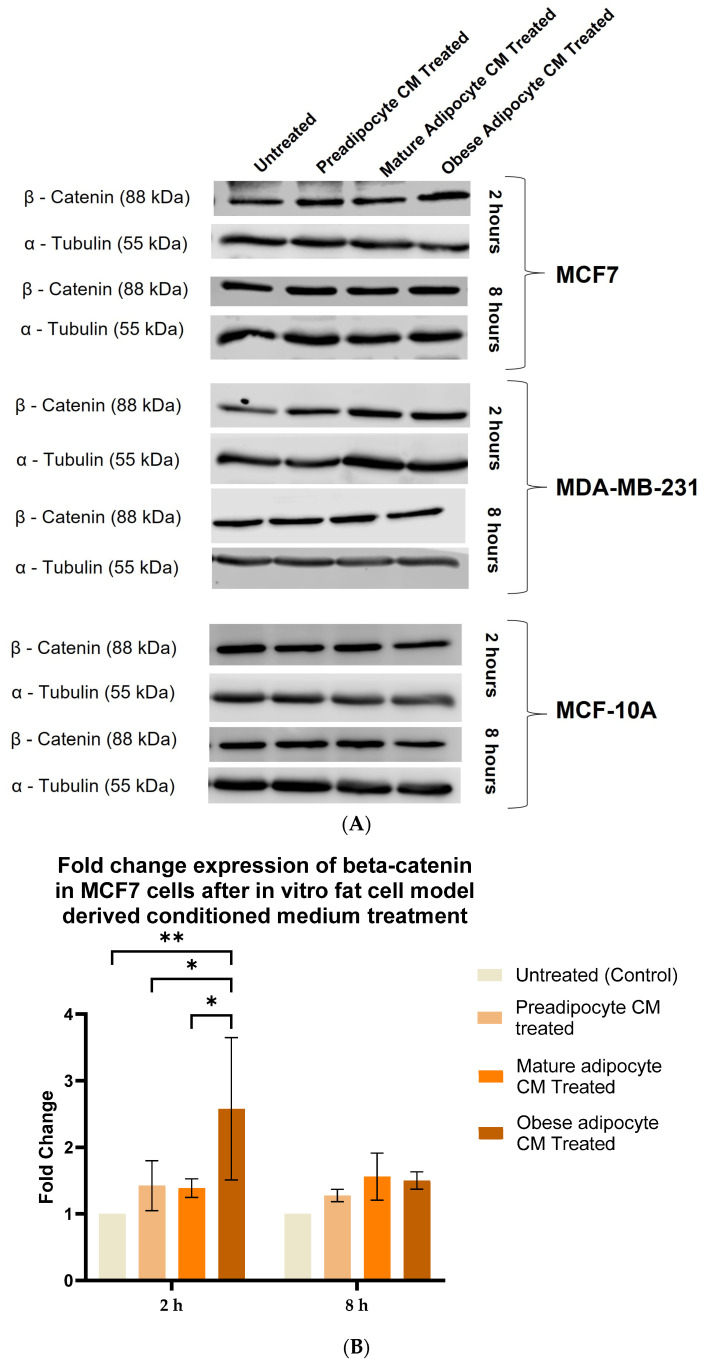
Adipocyte-secreted factors, under obese conditions in vitro, elevates beta-catenin stabilization. MCF7, MDA-MB-231 and MCF-10A, were exposed to in vitro fat cell-derived conditioned medium treatment for 2 and 8 h. Immunoblotting of total lysates of each cell type under each treatment condition was performed using specific antibodies. (**A**). Representative Western blot showing β-catenin and α-tubulin protein expression in breast cells detected with their respective antibodies. The blot signals for β-catenin on the membrane were normalized by densitometric analysis with respect to the loading control (α-tubulin) corresponding to each treatment. The bar graph represents fold change expression of β-catenin in mean ± SEM for MCF7 (**B**), MDA-MB-231 (**C**), and MCF-10A (**D**) when exposed to in vitro fat cell-derived conditioned medium treatment, respectively. The fold change expression was derived by dividing the normalized expression of β-catenin protein for each treated condition with respect to the normalized expression of β-catenin protein for untreated conditions for each cell line. The mean value of the fold change expression of β-catenin protein for each condition was derived from at least 3 independent experiments. The significance for the changes in β-catenin expression among the treated conditions, compared to untreated, for each cell line was measured using one-way ANOVA followed by Tukey’s post hoc test for multiple comparison. *p* value < 0.05 was considered statistically significant. One asterisk (*) indicates *p* ≤ 0.05, and two asterisks (**) indicate *p* ≤ 0.01. *p*-value > 0.05 was considered statistically nonsignificant (ns).

**Figure 5 ijms-24-16605-f005:**
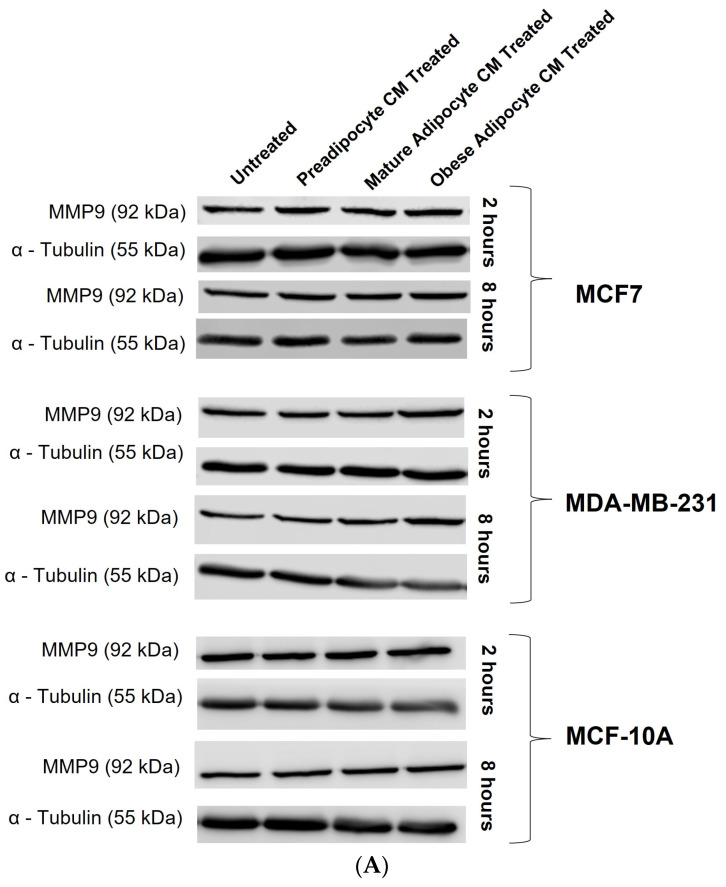
Effect of adipocyte-secreted factors, under obese conditions in vitro, elevates MMP9 expression. MCF7, MDA-MB-231, and MCF-10A were exposed to in vitro fat cell-derived conditioned medium treatments for 2 and 8 h. Total lysates of each cell type under each treatment condition were analyzed by Western blotting using specific antibodies. (**A**) Representative Western blotting shows MMP-9 and α-tubulin protein expression in breast cells detected with their respective antibodies. The protein signal for MMP-9 was normalized by densitometric analysis with respect to the signal of the loading control (α-tubulin) corresponding to each treatment. The bar graph represents fold change expression of MMP-9 in mean ± SEM for MCF7 (**B**), MDA-MB-231 (**C**), and MCF-10A (**D**) when exposed to in vitro fat cell-derived conditioned media treatments, respectively. The fold change expression was derived by dividing the normalized expression of MMP-9 protein for each treated condition with respect to the normalized expression of MMP-9 protein for untreated conditions for each cell line. The mean value of the fold change expression of MMP-9 protein for each condition was derived from at least 3 independent experiments. The significance for the changes in MMP-9 expression among the treated conditions, compared to the untreated, for each cell line was measured using one-way ANOVA followed by Tukey’s post hoc test for multiple comparison. *p* value < 0.05 was considered statistically significant. One asterisk (*) indicates *p* ≤ 0.05, and two asterisks (**) indicate *p* ≤ 0.01. *p*-value > 0.05 was considered statistically nonsignificant (ns).

**Figure 6 ijms-24-16605-f006:**
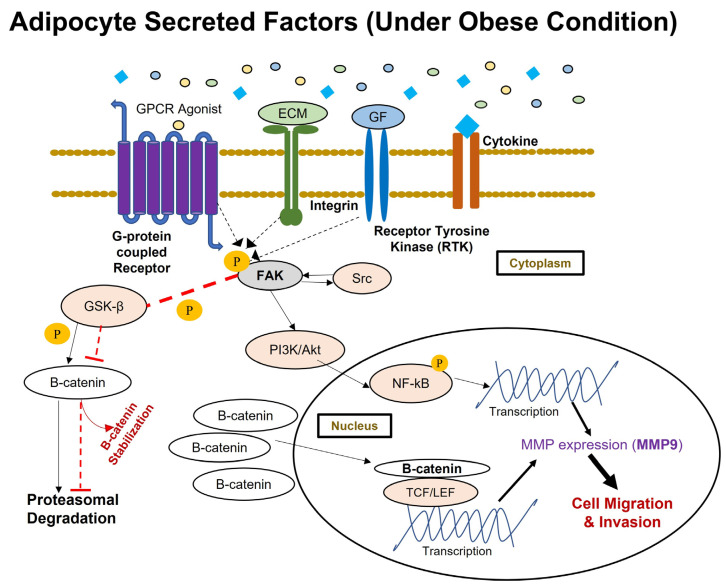
The proposed activation of FAK-mediated metastatic signal transduction pathway implicated in breast cancer cells in response to aberrant adipokine secretion under in vitro conditions mimicking obesity. FAK is activated through phosphorylation when extracellular stimuli in the microenvironment interact with integrin, receptor tyrosine kinase, and G-protein coupled receptors. This study has observed the aberrant secretion of adipokines under conditions mimicking obesity causing greater activation of FAK through its autophosphorylation at Tyr397 residue. Upon FAK activation, FAK-Src complexes will activate several downstream targets via phosphorylation. Activated FAK will cause proteasomal degradation of GSK3β through its phosphorylation, and for that reason, this study has observed an increased presence of β-catenin protein in breast cancer cells under in vitro conditions mimicking obesity. Stabilized β-catenin will then translocate to the nucleus and transcriptionally regulate the expression of several proteins associated with survival and metastatic events in breast cancer cells. This study has also observed enhanced expression of MMP-9 protein in breast cancer cells under in vitro conditions mimicking obesity, downstream to p-FAK through the PI3K/Akt/NF-kB signaling axis.

## Data Availability

Data is contained within the article.

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
