# Peer review of "The Effect of Adipocyte-Secreted Factors in Activating Focal Adhesion Kinase-Mediated Cell Signaling Pathway towards Metastasis in Breast Cancer Cells"

_ijms, 2023, doi:10.3390/ijms242316605_

Round 1
Reviewer 1 Report
Comments and Suggestions for Authors
Reviewer comments
The manuscript of the Noshin group for the first time explores the adipocyte-secreted factors on promoting migration/invasion through 2 activating fak in adipose tissue to evaluate whether this tissue can be considered to provide some sort of immunological sanctuary. The authors clearly illustrate that several inflammatory genes are upregulated upon metastatic progression of cancer. Using three different MCF7 16 (ER+), MDA-MB-231 (Triple Negative) breast cancer cells and MCF-10A non-tumorigenic breast 17 cells. authors document that increased FAK autophosphorylation was followed by increased expression of beta-catenin and MMP9 in the breast cancer cells when exposed to in vitro obese adipocyte conditioned medium, but not the MCF10A cells The manuscript is well written and the data clearly presented. A number of comments are included below to be considered for a revised version of the manuscript.
1. The structure of the article is complete, and the data is substantial.
2. The methods need to be supported by citing articles. Some sentences which are indicated in the manuscript need to be supported by reference.
3. Include statistical tool name in the method section
4. The introduction is very succinct and would benefit some extension to introduce the various immune parameters analyzed in the manuscript that are known to be relevant breast cancer.
Author Response
Reviewer Comments 1:
The manuscript of the Gollahon group for the first time, explores the adipocyte-secreted factors on promoting migration/invasion through 2 activating fak in adipose tissue to evaluate whether this tissue can be considered to provide some sort of immunological sanctuary. The authors clearly illustrate that several inflammatory genes are upregulated upon metastatic progression of cancer. Using three different MCF7 16 (ER+), MDA-MB-231 (Triple Negative) breast cancer cells and MCF-10A non-tumorigenic breast 17 cells. authors document that increased FAK autophosphorylation was followed by increased expression of beta-catenin and MMP9 in the breast cancer cells when exposed to in vitro obese adipocyte conditioned medium, but not the MCF10A cells. The manuscript is well written, and the data clearly presented. A number of comments are included below to be considered for a revised version of the manuscript.
- The structure of the article is complete, and the data is substantial.
The authors appreciate the positive comment.
- The methods need to be supported by citing articles. Some sentences which are indicated in the manuscript need to be supported by reference.
Response: The methods described within the manuscript were optimized by the authors from combining diverse protocols available online, and none of them were followed 100%. As a result, the methods have been written with all the necessary details to be recapitulated.
- Include statistical tool name in the method section
Response: The authors have stated in the “Statistical Analysis” portion of the “Method” that GraphPad Prism 9 software was used and the statistical analyses applied were described in the Statistical Analysis section of the Methods.
- The introduction is very succinct and would benefit some extension to introduce the various immune parameters analyzed in the manuscript that are known to be relevant breast cancer.
Response: The authors appreciate this suggestion. The discussion for the immune parameters relevant to this study has been discussed at length in prior work from their lab. “Characterizing 3T3-L1 MBX Adipocyte Cell Differentiation Maintained with Fatty Acids as an In Vitro Model to Study the Effects of Obesity”. This has been cited in the introduction for the benefit of the readers.
Reviewer 2 Report
Comments and Suggestions for Authors
The study targeted obesity-induced changes /secretory factors released by white adipocytes and their roles in the regulation of FAK in breast cancer cells. Data indicated that the secreted medium stimulated FAK activation, promoting breast cancer cell migration and invasion. The role of Fak has been explored in numerous studies. The manuscript requires a lot of work. English editing is required. There are other issues to address:
1. Title is overloaded and does not provide a clear logical statement. Abbreviations in the title should be avoided.
2. Abstract requires a lot of work. Some sentences are too wordy and should be divided in half ( at least). The last sentence -is too general and does not describe the main novelty of this work.
3. Introduction is too long and should be focused on the substances secreted from adipocytes as a main factors which links obesity to the activation of FAK. For instance, this study https://pubmed.ncbi.nlm.nih.gov/32859692/ was not cited or discussed. Possible involvement of PPAR was not mentioned.
4. Figure 1 is not relevant to the Introduction and should be presented in the Discussion after the data is analysed. It is a signaling pathway which should be presented after the data is discussed.
5. Results: authors need to do mass spectrometry and identify the secreted factors required for activation of FAK. That is main problem of this study.
6. Authors cannot claim that “ adipocytokines from in vitro adipocytes mimicking obesity, increased the stabilization of β-catenin and the expression of MMP-9 protein in breast cancer cells MCF-7 and MDA-MB-231”, as they have not tested the presence of adipokines. They did not identify any substances in the conditioned medium. Therefore, this conclusion cannot be used.
7. Discussion should be edited by a profession English language editor.
Comments on the Quality of English Language
Extensive English editing is required.
Author Response
Reviewer Comments 2:
The Authors appreciate the detailed review provided. Our responses are found below.
The study targeted obesity-induced changes /secretory factors released by white adipocytes and their roles in the regulation of FAK in breast cancer cells. Data indicated that the secreted medium stimulated FAK activation, promoting breast cancer cell migration and invasion. The role of Fak has been explored in numerous studies. The manuscript requires a lot of work. English editing is required. There are other issues to address:
- The title is overloaded and does not provide a clear logical statement. Abbreviations in the title should be avoided.
Response: The authors have suggested a more succinct title in the revised manuscript without any abbreviations.
- Abstract requires a lot of work. Some sentences are too wordy and should be divided in half (at least). The last sentence -is too general and does not describe the main novelty of this work.
Response: The authors have revised.
- Introduction is too long and should be focused on the substances secreted from adipocytes as a main factors which links obesity to the activation of FAK. For instance, this study https://pubmed.ncbi.nlm.nih.gov/32859692/ was not cited or discussed. Possible involvement of PPAR was not mentioned.
Response: The authors has discussed adipocyte secreted factors in a previous review article: Mubtasim N, Moustaid-Moussa N, Gollahon L. The Complex Biology of the Obesity-Induced, Metastasis-Promoting Tumor Microenvironment in Breast Cancer. Int J Mol Sci. 2022;23(5), and a research article: Mubtasim N, Gollahon L. Characterizing 3T3-L1 MBX Adipocyte Cell Differentiation Maintained with Fatty Acids as an In Vitro Model to Study the Effects of Obesity. Life. 2023;13(8):1712.
These citations have been moved from other sections of the manuscript, forward to the introduction where appropriate. The suggested article on FAK- and PPAR was not included as the authors’ felt that it was not relevant to this current study since it was covered in the citations listed above in addition to FAK activation, MMP9 and beta-catenin as the crux of this study.
- Figure 1 is not relevant to the Introduction and should be presented in the Discussion after the data is analyzed. It is a signaling pathway which should be presented after the data is discussed.
Response: The figure has been moved to the discussion.
- Results: authors need to do mass spectrometry and identify the secreted factors required for activation of FAK. That is main problem of this study.
Response: The authors have analyzed the adipocyte secreted factors that they have used as in vitro fat cell model using the R & D mouse adipokine array kit. The study results were discussed in the previous study: Characterizing 3T3-L1 MBX Adipocyte Cell Differentiation Maintained with Fatty Acids as an In Vitro Model to Study the Effects of Obesity. Life. 2023;13(8):1712 and cited accordingly. The authors have also disclosed the kit in the revised manuscript (Line 526-528).
- Authors cannot claim that “adipocytokines from in vitro adipocytes mimicking obesity, increased the stabilization of β-catenin and the expression of MMP-9 protein in breast cancer cells MCF-7 and MDA-MB-231”, as they have not tested the presence of adipokines. They did not identify any substances in the conditioned medium. Therefore, this conclusion cannot be used.
Response: Based on a related, previous study, (Characterizing 3T3-L1 MBX Adipocyte Cell Differentiation Maintained with Fatty Acids as an In Vitro Model to Study the Effects of Obesity. Life. 2023;13(8):1712) now cited, the authors analyzed the adipocyte secreted factors that they have used as in vitro fat cell model. The authors have rewritten this in context to more strongly emphasize these prior results, such that these adipokines have been identified and therefore, the conclusion is relevant to the results.
- Discussion should be edited by a profession English language editor.
Response: Without specific examples of poorly worded English, it is difficult for the corresponding author to identify areas of concern from the reviewer. As English is my first language, I have attempted to correct any issues the reviewer may have had with the writing.
Round 2
Reviewer 2 Report
Comments and Suggestions for Authors
Authors submitted the revised manuscript, but did not highlight the changed phrases. I request the resubmission of the revised version with ALL changes/additions highlighted.
Comments on the Quality of English Languageminor editing is required. For instance, lines 378-381; "The study reported how the adipocytokine secretion from the in vitro 3T3-L1 MBX cell derived adipocytes, in their different adipogenic forms, preadipocyte, mature adipocyte and obese adipocyte (adipocyte under condition mimicking obesity) [72], play a role in shaping the behavior of breast cells with respect to migration and invasion." This phrase is too long and unclear. Authors should break this phrase into 3 sentences. It is recommended to use concise language. It is wrong to write ' in vitro adipocytes mimicking obesity". what does it mean? did you mean " a model of obesity in adipocytes in vitro" ( line 418)? Many long sentences should be edited. Authors should avoid writing long sentences as the meaning of the content is getting lost.